# Prognostic Implications of the Residual Tumor Microenvironment after Neoadjuvant Chemotherapy in Triple-Negative Breast Cancer Patients without Pathological Complete Response

**DOI:** 10.3390/cancers15030597

**Published:** 2023-01-18

**Authors:** Marylène Lejeune, Laia Reverté, Esther Sauras, Noèlia Gallardo, Ramon Bosch, Albert Roso, Anna Petit, Vicente Peg, Francisco Riu, Joan García-Fontgivell, José Ibáñez, Fernanda Relea, Begoña Vieites, Catherine Bor, Luis de la Cruz-Merino, Meritxell Arenas, Valerie Rodriguez, Juana Galera, Anna Korzynska, Philippe Belhomme, Benoît Plancoulaine, Tomás Álvaro, Carlos López

**Affiliations:** 1Oncological Pathology and Bioinformatics Research Group, Molecular Biology and Research Section, Pathology Department, Hospital de Tortosa Verge de la Cinta, IISPV, URV, 43500 Tortosa, Spain; 2Clinical Studies Unit, Hospital de Tortosa Verge de la Cinta, Carretera Esplanetes, 14, 43500 Tortosa, Spain; 3Institut Universitari d’Investigació en Atenció Primària Jordi Gol (IDIAP Jordi Gol), Gran Via Corts Catalanes, 587, 08007 Barcelona, Spain; 4Pathology Department, Hospital Universitari de Bellvitge, 08907 Barcelona, Spain; 5Pathology Department, Hospital Universitari de Vall Hebron, 08035 Barcelona, Spain; 6Pathology Department, Hospital Universitari Sant Joan de Reus, 43204 Reus, Spain; 7Pathology Department, Hospital Universitari Joan XXIII, IISPV, 43005 Tarragona, Spain; 8Pathology Department, Hospital Universitario Virgen Macarena, 41009 Seville, Spain; 9Pathology Department, Hospital General de Ciudad Real, 13005 Ciudad Real, Spain; 10Pathology Department, Hospital Universitario Virgen del Rocío, 41013 Seville, Spain; 11Path-Image/BioTiCla, University of Caen, François Baclesse Comprehensive Cancer Center, 14000 Caen, France; 12Oncology Department, Hospital Universitario Virgen Macarena, 41009 Seville, Spain; 13Radiation Oncology Department, Hospital Universitari Sant Joan de Reus, IISPV, Universitat Rovira I Virgili, 43204 Reus, Spain; 14Oncology Department, Hospital de Tortosa Verge de la Cinta, IISPV, 43500 Tortosa, Spain; 15Gynaecology Department, Hospital Universitari Joan XXIII, IISPV, 43005 Tarragona, Spain; 16Laboratory of Processing Systems of Microscopic Image Information, Nalecz Institute of Biocybernetics and Biomedical Engineering, Polish Academy of Sciences, 02-109 Warsaw, Poland; 17ANTICIPE, INSERM, François Baclesse Comprehensive Cancer Center, University Caen Normandy, 14000 Caen, France

**Keywords:** triple-negative breast cancer, neoadjuvant therapy, relapse, survival, tumor microenvironment, genetic markers, immune markers

## Abstract

**Simple Summary:**

Triple-negative breast cancer (TNBC) is currently in the clinical research spotlight because of the tumor’s aggressive and invasive nature and the scarcity of therapeutic targets. Despite recent advances in identifying reliable prognostic biomarkers in the tumor microenvironment (TME), rigorous evaluation of their predictive capacity remains challenging. We describe the immune cellular and genetic profile of the residual tumor of TNBC that does not achieve a pathological complete response (pCR) after neoadjuvant chemotherapy (NAC). A high concentration of lymphocytes and dendritic cells, as well as genetic TME markers such as MUC-1 and CXCL13 in the residual tumor, are valuable prognostic factors of survival and relapse in TNBC patients. From a clinical health perspective, a thorough understanding of the composition of the TME and its prognostic implications might yield relevant immunological information and reveal key predictive biomarkers. This could ultimately help substantially improve the outcomes of residual cancer-burdened TNBC patients after NAC.

**Abstract:**

With a high risk of relapse and death, and a poor or absent response to therapeutics, the triple-negative breast cancer (TNBC) subtype is particularly challenging, especially in patients who cannot achieve a pathological complete response (pCR) after neoadjuvant chemotherapy (NAC). Although the tumor microenvironment (TME) is known to influence disease progression and the effectiveness of therapeutics, its predictive and prognostic potential remains uncertain. This work aimed to define the residual TME profile after NAC of a retrospective cohort with 96 TNBC patients by immunohistochemical staining (cell markers) and chromogenic in situ hybridization (genetic markers). Kaplan–Meier curves were used to estimate the influence of the selected TME markers on five-year overall survival (OS) and relapse-free survival (RFS) probabilities. The risks of each variable being associated with relapse and death were determined through univariate and multivariate Cox analyses. We describe a unique tumor-infiltrating immune profile with high levels of lymphocytes (CD4, FOXP3) and dendritic cells (CD21, CD1a and CD83) that are valuable prognostic factors in post-NAC TNBC patients. Our study also demonstrates the value of considering not only cellular but also genetic TME markers such as MUC-1 and CXCL13 in routine clinical diagnosis to refine prognosis modelling.

## 1. Introduction

Triple-negative breast cancer (TNBC), which accounts for 10–20% of all breast cancers (BCs) worldwide [1], is characterized by the lack of estrogen receptor (ER), progesterone receptor (PR) and human epidermal growth factor receptor 2 (HER2) expression [2]. Compared with the other subtypes, TNBC has an aggressive nature, a higher metastatic potential, and a tendency towards a worse prognosis and higher risks of local/distant relapse and death 3–5 years after diagnosis. It is not sensitive to endocrine therapy or HER2 treatment, so there is a need to improve standardized TNBC treatment regimens [3]. Current standard-of-care treatments for early-stage TNBC consists of preoperative neoadjuvant chemotherapy (NAC) followed by surgery and radiotherapy [4], a breast-conserving treatment that allows the tumor response to be evaluated in vivo and clinical decisions to be taken in situ based on treatment effectiveness [5]. The tumor response to NAC is a strong predictive factor of patient outcome and prognosis [6] and is evaluated by means of pathological complete response (pCR) rates. A pCR after NAC in TNBC represents a surrogate marker endpoint for a prediction of good prognosis [7,8], but some patients are able to achieve long-term survival without relapse even though they do not attain a pCR [9]. There is no clear explanation for such behavior.

Besides the type of pathological response to NAC, the tumor microenvironment (TME) has been recognized as an important modulator of carcinogenesis, whereby it is likely to play a pivotal role in tumor development and disease progression [10]. The TME is composed of a diversity of cell subtypes, ranging from immunological to stromal cells, including extracellular matrix (ECM) and soluble factors. There is a complex interplay between the cancer cells and the TME components at the tumor site that influences overall therapy effectiveness and clinical outcomes. Moreover, specific therapies and cytotoxic agents have been shown to alter and reorganize the ECM and modulate stromal elements such as the immune response of the residual TME. In this context, immune cells are one of the main players within the TME, and many studies have tried to characterize them with respect to the BC subtype in the pre-NAC [11,12,13] and post-NAC settings [13,14,15]. Of all the immune infiltrates, tumor-infiltrating lymphocytes (TILs) are of particular concern since their levels are positively correlated with longer disease-free survival (DFS) and improved survival rates in TNBC patients when evaluated in pre-NAC biopsies [14,16,17,18,19,20,21]. Although several TME elements before NAC have been explored in depth, relatively few studies have addressed them in the post-NAC context [19,22,23,24], most efforts having focused on determining the predictive role of the biomarkers in response to NAC rather than on BC prognosis [25,26]. Along with the presence of immune TME cells, the expression of other stromal components such as immunological genes is also known to modulate the clinical course of TNBC patients, although this has not been extensively studied in the pre-NAC and/or post-NAC contexts [27,28,29,30]. Indeed, emerging research has centered its attention on developing new immune signatures of biomarkers in TNBC that can predict patient prognosis, rather than evaluating the consequences of them having received NAC or exploring the benefit of immunotherapy [31,32]. To our knowledge, very few reports that focus on profiling the residual tumor nest in the neoadjuvant setting in TNBC have included both cellular and genetic elements [33]. Therefore, to improve our knowledge about how the residual TME contributes to TNBC progression, we consider it important to understand the composition of the entire tumor microenvironment and to determine the prognostic implications of the most critical constituents.

For this purpose, the present study aims to provide a broader view of the TME components, including not only immune and stromal cells, but also the mRNA expression levels of certain cytokines, interleukins (ILs) and matrix metalloproteases (MMPs) of the residual tumor nest of TNBC patients with no or partial pCR after NAC. The work also explores the prognostic value of the selected biomarkers, alone and in combination, for predicting the proneness to relapse and survival of TNBC patients.

## 2. Design and Methods

### 2.1. Study Design and Target Population

This retrospective cohort study involved 96 TNBC patients diagnosed with invasive breast carcinoma of no special type (NST) between 2008 and 2013. The TNBC diagnosis had been confirmed with a core needle biopsy before NAC in accordance with the WHO classification [2]. To be included in the study, patients could not have had distant metastases at the time of diagnosis, or a pCR after treatment with NAC (no response or partial response) and had to have undergone breast-conserving surgery or had a mastectomy. A pCR in the neoadjuvant setting is defined as the absence of residual invasive cancer in the surgical specimen and in the ipsilateral/regional lymph nodes in the sample obtained after completion of therapy [34]. The pathological response was assessed with respect to the Miller–Payne [35] and residual cancer burden (RCB) [36] grading systems.

All post-NAC biopsies of TNBC patients selected for the study were obtained from cancer biobanks or tumor banks of the pathology services of the participating centers (Hospital de Tortosa Verge de la Cinta, Hospital Universitari de Bellvitge, Hospital Universitari de Vall Hebron, Hospital Universitari Sant Joan de Reus, Hospital Universitari Joan XXIII, Hospital Universitario Virgen Macarena, Hospital General de Ciudad Real, Hospital Universitario Virgen del Rocío, Hospital Universitario Virgen Macarena), having obtained their informed consent. Patients were followed up for as long as 5 years from the date of surgery.

The primary study endpoints included survival and relapse status, overall survival (OS) and relapse-free survival (RFS) rates. Survival status refers to patients being alive or dead at the end of follow-up. Relapse status is defined with respect to the return of the disease after a period of improvement (presence or absence). OS represents the period between surgery date and death by any cause, or the date of last follow-up. RFS is the interval between surgery date and the date of relapse or death. Patients who did not experience the event were censored at the date of their last follow-up (60 months).

Clinical/pathological characteristics and TME markers were also studied. The clinical/pathological data collected from patients at diagnosis and during follow-up were age at diagnosis, histological grade, TNM stage, tumor diameter, menopausal status, nodal status, Ki67, type of treatment/surgery, death (if this occurred) and the type of response (partial or absent) to NAC. The TME markers studied by immunohistochemistry (IHC) were as follows: CD4^+^ and CD8^+^ T lymphocytes, FOXP3^+^ regulatory T cells, CD68^+^ macrophages, CD57^+^ natural killer (NK) cells, CD1a^+^ Langerhans dendritic cells (DCs), CD21^+^ follicular DCs and CD83^+^ mature DCs, CD15^+^ granulocytes, CD31^+^ endothelial cells, CD34^+^ immature myeloid cells (iMCs), HLA-DR^+^ antigen-presenting cell (APC) markers and the cell surface heparin sulfate proteoglycan Syndecan-1 (Synd1, CD138). The expression of the mRNA of cytokines and interleukins was studied by chromogenic in situ hybridization (ISH) for the MMPs associated with ECM remodeling (MMP1, MMP-9, MMP12); the immunoregulatory cytokines (IL6, IL10, IL15, TNF-α), the C-X-C motif chemokine ligand 13 (CXCL13) and the tumor-associated epithelial oncoprotein mucin-1 (MUC1). All these variables were compared in the different groups of the TNBC patient cohort with respect to their survival status, relapse status and the type of response to NAC (partial or no response). The choice of markers to evaluate was based on their immunological relevance previously demonstrated in BC studies, mostly in the neoadjuvant setting. The markers were as follows: CD4, CD8, FOXP3, CD1a, CD68, CD83 [16], CD57, CD15 [37], MUC1 [38], MMP-9 [28], CXCL13, IL15, FOXP3, HLA-DR [27], IL-6, IL-10, TNFα [39], CD21 [40], all MMPs [41], CD31 [42], CD34 [43] and CD138 [44].

### 2.2. TMA Preparation for the Detection of TME Markers

Two representative 2-mm diameter cylinders from the residual tumor biopsy site were extracted from the surgical specimen removed post-NAC in each confirmed TNBC case in the study. The exact position of these cylinders corresponds to the representative areas of the residual tumor, with TME components selected by pathologists from corresponding slides of the biopsy stained with hematoxylin-eosin. A total of 192 cylinders obtained from the 96 cases in the study were transferred to a paraffin mold with the Arraymold tissue microarray (TMA) tool. Each mold comprises 50 holes, so four TMAs were required for full coverage. Although correspondence between TMAs and whole-tissue sections may not be ideal at the diagnostic level for individual patients, it is considered technically appropriate for high-throughput sample analysis [45] and has been widely used in BC research for to evaluate TME biomarkers [46]. After TMA construction, 22 sections of 2 µm were made to stain the different markers. IHC was performed by the ENDVISION FLEX™ (Santa Clara, CA, USA) method using diaminobenzidine chromogen (DAB) as the substrate, according to the supplier’s instructions and laboratory protocol. The antibodies used in the study, with their source and dilutions, are presented in Appendix A.

Tissue ISH assays, using the ViewRNA QuantiGene^®^ kit (Affymetrix, Inc., Santa Clara, CA, USA), were performed by Sophistolab (Sophilstolab AG, Muttenz, Switzerland) to detect the mRNA expression levels of the cytokine and stromal markers. The probes used are listed in Appendix A. These ISH assays enable the specific location of nucleic acid targets in fixed tissues with virtually no background to be determined. The signal is detected using Fast Red chromogen substrate. The presence of positive staining of mRNA is visualized by the presence of red dots.

### 2.3. Slide-Scanning and Digital Image Analysis

IHC- and ISH-stained slides of each TMA were digitized with an Aperio ScanScope CS slide Scanner at 20X/NA 0.75 to produce LZW lossless compression profiles in SVS format. Subsequently, each cylinder of the scanned TMAs was extracted with specific automated algorithms developed by the Path-Image/BioTiCla research group (Appendix A) and saved as an individual image in TIFF format. This procedure produced 4224 images (192 cylinders × 22 stains) [47]. A total of 2496 digital images of IHC-stained markers were evaluated by an automated digital image analysis procedure established in previous studies [48,49] using FIJI software [50]. The algorithms were adjusted to suit the specific markers used in the study and validated by expert pathologists and biologists. These procedures quantify the percentage of positive-stained areas of each marker relative to the whole area of the cylinder, all measurements being expressed in pixels. The final results for each stained marker are the mean percentages of positive pixels of the two cylinders studied in each patient. In cases where one of the cylinders was lost due to poor staining, loss of tissue or poor image quality (between 3.1% and 9.4% of cases for each marker), we used information from a single cylinder.

The other 1728 digital images of stained IHS mRNA markers were evaluated manually by two independent observers and classified as absence of mRNA expression if there was no staining or if there were fewer than 10% of red dots, and classified as presence of mRNA expression for cylinders with ≥10% red dots. The highest value of the two cylinders studied in each patient was chosen as the final result. A cut-off value of 10% was selected based on the threshold of positive staining observed in our cohort that best distinguished the groups of patients.

### 2.4. Statistical Analysis

Differences in the median concentrations of the IHC-stained markers were determined in biopsies post-NAC between groups of patients (alive vs. dead, relapsed vs. non-relapsed) using unpaired samples *t*-tests for normally distributed data, or Mann–Whitney U tests for non-normally distributed data, normality first being assessed using the Kolmogorov–Smirnov test. Differences in the percentage of patients expressing mRNA of markers detected with ISH assays were evaluated in biopsies post-NAC for the patient groups (alive vs. dead, relapsed vs. non-relapsed) using the chi-square or Fisher exact test, as appropriate.

The Kaplan–Meier (K–M) method was used to estimate the OS and RFS among all patients for each marker studied (determined by IHC and ISH). Superimposed K–M curves were derived for each studied marker to compare markers with concentrations above (high) and below (low) the median for IHC markers and between those markers expressing or not expressing mRNA for IHS markers, stratified by the patient’s nodal status (positive or negative) at diagnosis and their response (none or partial) to NAC. Biomarkers that were significant in at least two of the previous comparison sets (stratified or not by nodal status at diagnosis, and by the type of response to NAC) were selected for subsequent analysis. The combinations of these selected biomarkers with a concentration greater than the median were used to categorize patients into six groups (from 0 to 5) according to the number of the significant biomarkers they expressed. This categorization was used to construct K–M curves to further evaluate the individual and combined capacity of the selected biomarkers and thereby estimate the OS and RFS of all patients.

Univariate and multivariate Cox regression analyses were conducted to estimate the risk of relapse and death associated with each variable in terms of hazard ratios (HRs) and their associated 95% confidence intervals (CIs). Markers with a significance of *p* < 0.10 in the univariate Cox models were evaluated in the final multivariate models. The final models included all variables with a level of significance of *p* < 0.05, thereby creating a predictive model for patient relapse and death. The models obtained were internally validated through the bootstrapping simulation technique using IBM SPSS Statistics for Windows version 23.0 (IBM Corp., Armonk, NY, USA). Means and 95% Cis were estimated from the 10,000 bootstrapped data sets for each scenario. The ability of the final multivariate regression models to predict the likelihood of death and relapse before and after bootstrapping validation was assessed by considering the sensitivity, specificity and area under the curve (AUC) of the receiver operating characteristic (ROC) curves. Statistical analyses were performed with IBM SPSS for Windows version 21.0 (IBM Corp., Armonk, NY, USA) and Stata software version 14.0 (StataCorp LLC, College Station, TX, USA).

### 2.5. Ethical Considerations

The project was approved by the Internal Scientific Committee of the Hospital de Tortosa Verge de la Cinta and the Ethics Committee for Clinical Research (CEIC) of Tarragona on 8 November 2013 (Ref: 55p/2013). We followed the guidelines provided by the Strengthening the Reporting of Observational Studies in Epidemiology (STROBE). The treatment, communication and transfer of personal data of all participants complied with Spanish Law 15/1999 and the European Directive 95/46/EC concerning the Protection of Personal Data. The data generated and collected during the study were held in a separate database in accordance with the dissociation procedure to ensure the invulnerability of information and the confidentiality and privacy of patients.

## 3. Results

### 3.1. Clinico-Pathological, Cellular and Genetic Characteristics of the Patient Cohort

Considering the entire study population, most TNBC patients presented similar tumor features at diagnosis (Table 1): a high degree of Ki67 (>30%), histological grade III, partial response to NAC, and submitted to mastectomy rather than tumorectomy followed by lymphadenectomy. After surgery, most patients did not receive chemotherapy but did require radiotherapy.

Patients who relapsed or died during follow-up presented a tumor with a significantly greater diameter and a higher percentage of positive nodal status at diagnosis than the other groups of patients. Moreover, 82.9% of the patients who relapsed died during the follow-up. Similarly, patients who died had lower relapse-free survival than patients who were alive at the end of follow-up.

Regarding the concentrations of markers quantified by IHC, patients who did not die or relapse during follow-up featured more immune cells: CD4^+^ T lymphocytes, CD8^+^ T lymphocytes, FOXP3^+^ regulatory T cells, CD1a^+^ and follicular CD21^+^ DC than the other groups (Table 2). The differences in the content of markers between groups were statistically significant or nearly significant (as in the cases of CD8^+^ and CD1a^+^). The non-significance of the latter marker was largely down to the small number of patients available. In the case of the mRNA expression levels of markers determined by ISH, patients who died during follow-up expressed higher MUC1 mRNA levels, and those who relapsed had lower CXCL13 mRNA levels, than the other groups. The differences were on the borderline of statistical significance.

### 3.2. Survival Curve Analysis According to IHC Markers and mRNA Expression Levels

Overall, of all the markers evaluated (Appendix A), only high levels of CD4^+^ T lymphocytes, FOXP3^+^ regulatory T cells, CD1a^+^ and CD21^+^ DC were significantly associated related to better OS (Figure 1A) and a lower risk of relapse (Figure 2A). High levels of CD83^+^ DC were also associated with a lower risk of relapse (Figure 2A).

Regarding the mRNA expression levels, patients who did not express MUC1 mRNA in the biopsy post-NAC had significantly better OS (*p* = 0.016) (Figure 1A), and those that expressed CXCL13 mRNA were at lower risk of relapse (*p* = 0.040) (Figure 2A), compared with the other groups.

It is striking that stratifying the concentrations of IHC markers by their nodal status revealed differences between the groups of patients (Figure 1B, Figure 2B, Appendix A). Patients with a positive nodal status at diagnosis and high concentrations of CD4^+^ lymphocytes, CD1a^+^ and CD21^+^ DC had better OS and RFS. On the other hand, patients with a negative nodal status and high concentration of FOXP3^+^ regulatory T cells had a significantly longer OS and RFS. Remarkably, although globally the high concentration of CD8^+^ lymphocytes was insufficient to identify patients with significantly better outcomes (Appendix A), stratifying patients by their nodal status identified a unique group with positive nodes and high CD8^+^ concentrations that had a significantly lower risk of relapse (Figure 2B). Concerning the levels of mRNA expression in markers evaluated by ISH, only the lack of MUC1 mRNA expression with a negative nodal status was related to improved survival rates (Figure 1B), while in no case was the expression of the ISH markers stratified by nodal status significantly related to the risk of relapse (Figure 2B).

Evaluating the relationship between markers concentrations and survival and relapse, stratifying patients in relation to their response to NAC (Appendix A) suggested that the content of the immune IHC markers more strongly affected patients with a partial response than it did those with no response (Figure 1C and Figure 2C). Accordingly, patients partially responding to NAC with high CD4, FOXP3 and CD83 contents had better OS and RFS than those with low concentrations. The only marker that showed statistically significant differences between the patient groups (partial responders and non-responders) was CD21, in whom high levels of this marker were related to a lower risk of relapse. Regarding the relationship between mRNA expression levels and outcomes by their response category, better OS was related to patients partially responding to NAC and with positive expression of IL-15, and with non-responders who did not express MUC1.

The combinatory effect of high levels of the selected immune markers (CD4, FOXP3, CD1a, CD21, CD83) on survival and relapse probability was evaluated further by considering the Kaplan–Meier curves (Figure 3A,B). Globally, these curves show how the presence of greater numbers of biomarkers present at concentrations above the median concentration provides evidence of a gradient in the increase in OS and RFS. Patients exhibiting five markers over the median were significantly different from those with only one or no markers. The survival probability reduced sharply from 100% to 36.36% and 40.00%, respectively, and the RFS probability dropped from 100% to 27.27% and 33.33%, respectively.

### 3.3. Univariate and Multivariate Cox Regression of Markers and Clinico-Pathological Findings in Residual Tumor Post-NAC

Variables and markers considered in the study were subsequently examined in univariate and multivariate Cox regression models to examine further the associations of each factor with the risk of relapse and death (Table 3).

Univariate regression of clinico-pathological variables identified large tumor diameter and positive nodal status at diagnosis as factors significantly associated with greater risks of relapse and death. Of the TME markers studied, univariate regression identified high levels of CD4^+^ and FOXP3^+^ T lymphocytes, HLA-DR^+^ APC and CD31^+^ endothelial cells to be associated with lower risks of relapse and death. With respect to the genetic markers of the TME, the presence of MUC1 mRNA expression and the deficiency in CXCL13 mRNA expression in residual tumors post-NAC significantly raised the risks of relapse and of death, respectively.

Multivariate Cox regression determined that positive nodal status at diagnosis and the presence of MUC1 mRNA expression in the residual tumor both remained as factors independently associated with a higher risk of death during the 5-year follow-up, and that raised CD4^+^ concentration continues to be a factor protecting against death. Each extra unit of CD4^+^ concentration in the residual tumor reduces the risk of death by 35.8%. The tumor diameter and concentrations of CD8^+^, FOXP3^+^, HLA-DR^+^ and CD31^+^ were no longer associated with poor survival rates. Bootstrap validation showed the HR indicating risk of death was lower with positive nodal status and slightly higher when MUC1 was expressed, the significant association of both with poor OS being retained. Conversely, CD4^+^ concentration was removed from the model after internal validation (*p* = 0.083). The ROC curves yielded similar AUCs for the multivariate model of OS before and after bootstrapping validation (0.7783 vs. 0.7746), poor sensitivity (58.82 vs. 50.46%) and good specificity (82.69%) (Figure 4A,B, respectively).

Turning to the RFS-related factors, the multivariate Cox regression analysis identified positive nodal status and large tumor diameter to be indicators of high risk of relapse, and the expression of CXCL13 mRNA as a factor protecting against relapse. The variables CD4, FOXP3, HLA-DR and CD31 were dropped from the model. Bootstrapping validation (Table 3) and the AUC values of the ROCs, and the sensitivity and specificity (Figure 4C,D) confirmed that it was appropriate to include these three variables in the model.

Finally, we compared the groups of patients with five markers over the median with the group with no markers concerning the variables significantly associated with the risk of death and/or relapse in the Cox regression analysis. While tumor diameter, nodal status and MUC-1 expression were similar in the two groups, the group exhibiting five immune markers contained higher HLA-DR contents (*p* < 0.0001) and CXCL13 expression levels (*p* = 0.001) than those with none of the selected markers.

## 4. Discussion

The immune infiltrates in the TME have been recognized as a heterogeneous system of varying clinical significance, exhibiting the highest infiltration rate in the TNBC phenotype. It is worth noting that the immunological cellular and molecular composition of the TME is likely to play a critical role in tumor development and progression [51]. In this context, biomarkers have emerged as powerful tools capable of predicting survival and recurrence probabilities or even the response to a therapeutic intervention [52]. Despite their clinical impact, the underlying diversity across immune TME elements depending on the disease stage in TNBC subtype has hampered the task of correctly assigning them a prognostic value, particularly in the neoadjuvant setting. Hence, the systematic determination of the quantity and the roles of reliable biomarkers integrating clinico-pathological information is highly desirable in order to facilitate BC management in routine clinical practice. Such biomarker identification might reveal mechanistic pathways beyond disease progression, thereby enabling improved prediction of TNBC diagnosis after NAC and demonstrate the efficacy of immunotherapy [53].

To shed light on this issue, we first determined the immune cell infiltrate and differential gene expression profile in the tumor nest of TNBC patients who were unable to develop a pCR after NAC. Following the TME content assessment, we investigated the predictive and prognostic value of the distinct TME cellular and genetic patterns in the disease outcomes (relapse and survival), taking into account the nodal status and type of response to NAC (no response vs. partial response).

Up to now, tumor diameter and nodal status at diagnosis have been among the best established prognostic factors for BC. In our cohort, patients with a larger tumor diameter and with positive nodal status presented a higher risk of relapse and death than the opposite groups. Whereas the two variables appeared related to RFS in the univariate and multivariate regression analyses, only the nodal status at the time of diagnosis turned out to be the most powerful prognostic factor in both final multivariate models (HR of 4.061 and 2.749 for OS and RFS, respectively). These results validated our cohort and reinforced the prognosis potential of tumor diameter and nodal status in daily routine BC diagnosis [54,55].

Along with nodal status and tumor size at diagnosis, immune cells, cytokines and certain stromal proteins infiltrating the breast tumor nest are decisive in achieving an effective anti-tumoral response, thereby ameliorating disease progression [30]. Among the lymphocyte populations studied within the TME post-NAC of our cohort, K-M analyses estimated that low levels of CD4^+^ and FOXP3^+^ cells are significantly related to decreased survival and increased likelihood of relapse in TNBC patients. However, only those patients with a positive nodal status along with low levels of CD4 were linked to a worse prognosis (shorter survival and higher relapse). Conversely, those patients with a negative nodal status along with a low FOXP3^+^ content presented a significant shorter survival and higher relapse. For instance, previous studies established that high levels of CD4^+^ cells affected long-term prognosis for TNBC, leading to better OS and DFS in those patients treated with NAC [22] and better OS and RFS when patients had positive nodal status [49,56]. Even with the prognostic value of classic lymphocytic subset markers in predicting OS and RFS being more clearly pinpointed in the TME, the varying prognostic role of FOXP3^+^ infiltrating cells in pre- or post-NAC TME of TNBC is still a matter of debate [18,21]. Our findings support the hypothesis proposing that cytotoxic agents such as those of the NAC switch the FOXP3 role in the TME from being anti-tumoral immunosuppressors to becoming effective anti-tumoral response promoters, especially in non-responders [57]. This confirms the relationship of high FOXP3, CD4 and CD8 levels with longer event-free survival (EFS) [58] and long-term prognosis for TNBC [59], and explains why NAC significantly reduces CD4 and FOXP3 T cell counts [60]. On the other hand, other authors have described low FOXP3 infiltrate as being a predictor of favorable tumor prognosis [61,62].

In addition to low CD4 and FOXP3 levels, Cox univariate but not multivariate analyses determined HLA-DR as predictive biomarkers of OS and RFS in our cohort. HLA-DR expression is essential to set up a suitable anti-tumor response by activating cytotoxic T lymphocytes (CTLs) and eventually inducing effector molecules’ release (such as IFN-γ, granzymes B, among others) to the tumor nest, which downregulation has been recognized as the main mechanisms of cancer immune escape [63]. In this context, Perez-Pena and colleagues found TNBC patients without a pCR expressing lower HLA-DR levels had earlier recurrences mostly with a fatal outcome than those with a pCR who did not relapse [64]. These results reinforced the crucial role of HLA molecules impairing tumour cells to evade immune system and spread elsewhere, thus, enhancing disease-free survival [65].

Regarding tumor-infiltrating DCs, in spite of being well-recognized as APC crucial for both innate and adaptive immune system, their contribution in the context of TNBC prognosis is largely unexplored, especially in the neoadjuvant setting. Considering the entire cohort regardless other clinico-pathological variables of interest, low levels of CD1a^+^ Langerhans DC, CD21^+^ follicular and mature CD83^+^ DC are significantly related to a higher risk to relapse, but only high levels of CD1a^+^ and CD21^+^ DC (not mature CD83^+^ DC) were found to improve OS. Moreover, these DCs play a distinguished role if stratifying patients by their nodal status and type of response. On the one hand, having a partial response after NAC is significantly correlated with high mature CD83^+^ DC counts with improved OS and DFS rates, high CD21 counts being associated with longer RFS, but not significantly linked to improve OS. On the other hand, patients with a positive nodal status along with high CD1a^+^ and CD21+ DCs levels in the nest of biopsies have a longer RFS and OS. These latter results appear to contradict those reporting the downregulation of CD1a^+^ DC gene in relation to BC patients achieving pCR after NAC [66], but were in line with authors associating the presence of high CD1a^+^ counts with longer progression free-survival than patients with lesions with a low level of infiltration [67]. All in all, the above findings seem to be in line with our knowledge that NAC changes cellular content in the TME and that positive nodal status at diagnosis is a strong predictive factor of poor outcomes (higher recurrence and death risk and lower treatment efficacy). Accordingly, we surmise that the immune response of patients with positive nodal status does not seem to be as effective as that in patients without metastatic lymph nodes, so predominant existing DCs are probably immature in nature (CD1a and CD21), and will therefore have worse prognosis. Conversely, patients with a more effective anti-tumor response (mature CD83) would also be likely to respond to NAC (partial response vs. no response), resulting in better outcomes (lower risk of relapse, and longer OS). Nonetheless, research in this direction is necessary to better assign the specific prognostic implications of immature and mature DCs at the tumor site for predicting BC outcomes.

As clearly demonstrated here, ongoing research on BC is shifting the study paradigm from the cellular to the genetic perspective to depict the dynamic interactions between the neoplastic cells and the TME [68] as well as the impact of the molecular changes induced by the therapeutic treatments. Accordingly, the approach proposed herein, which combines cellular composition and molecular expression levels, has allowed us to identify robust biomarkers that might mirror a more realistic influence of the whole TME milieu on disease outcomes. This has been reflected in two ways. Firstly, we identified the expression of MUC1 mRNA together with low CD4^+^ levels and a positive nodal status as being the strongest independent factors predictive of poor OS. MUC1 is a transmembrane oncoprotein implicated in cell–cell and cell–extracellular matrix adhesion, which contributes to immune escape and has been linked to transformation, invasion and progress of tumor cells [69,70]. In line with our findings, expression of MUC1 has been associated with poor prognosis in BC patients [71,72]. Furthermore, K-M curves confirm the usefulness of the expression of MUC1 mRNA as a genetic TME marker related to a poor OS. Secondly, CXCL13 mRNA expression together with negative nodal status and small tumor diameter are the strongest independent factors associated with better RFS. CXCL13, also known as B Cell-Attracting chemokine 1 (BCA-1), is a chemokine involved in anti-tumor immune response and B-cell attraction in germinal centers. In accordance with our work, the high level of infiltration of CXCL13-producing CD4^+^ follicular helper T cells and a CXCL13 gene expression signature were related to longer disease-free survival in TNBC patients [27,73]. All in all, these results highlighted the relevance of the expression of chemokines such as CXCL13 and the lack of expression of the transmembrane oncoprotein MUC-1 as surrogate markers of the anti-tumor immune response mediated by activated lymphocytes in the TME. The results also suggested that implementation of these markers in daily routine diagnosis might potentially improve diagnosis in TNBC patients following neoadjuvant treatments.

Ultimately, a combinatorial analysis of five of the selected biomarkers (CD4, FOXP3, CD1a, CD21 and CD83) allowed us to categorize the study cohort into two well-differentiated groups with distinct clinical outcomes: high- and low-risk patients. The high-risk group stands out because of a poor anti-tumor response (HLA-DR and CXCL13 downregulation) and low immune infiltrate content of the five biomarkers in the TME, which result in immune system evasion, favor tumor growth and progression, and lead to poor survival and RFS rates. In contrast, the low-risk group comprises patients with highly immunogenic TME (high immune infiltrate content of the five biomarkers), and a high level of expression of the gene signature CXCL13 and HLA-DR, contributing to an effective anti-tumor response, preventing tumor cells from escaping the immune surveillance system, benefiting from immunotherapy, lowering the risk of recurrence and improving survival rates. In addition to the previously reported promising immune fingerprints [31,64], the assessment of genetic biomarkers in parallel with the classic TME markers appears to be a promising tool for further stratifying TNBC patients to better understand TNBC complexity and, especially, patients receiving NAC who are unable to achieve pCR. In future studies, not only the immune infiltrates present in the intratumoral area but also those in the peritumoral area will be taken into account to better address the heterogeneity within the TME of the TNBC phenotype.

## 5. Conclusions

Our study demonstrates the combinatorial effect of elevated levels of T lymphocytes (CD4 and FOXP3) and DC (CD21, Cd1a and CD83) in the TME being a helpful marker for predicting more favorable outcomes of patients who have not had a pCR post-NAC (lower recurrence risk and higher survival rates). In addition, along with well-established clinico-pathological factors (positive nodal status and large tumor diameter), the lack of CXCL13 expression and the expression of MUC1 in the tumor bed were identified as risk factors that are strongly associated with worse RFS and OS, respectively. Overall, we conclusively demonstrate the benefit of combining cellular and molecular markers in the search for potential biomarkers to predict BC prognosis in TNBC patients with residual cancer burden after NAC. It is worth noting that, following external validation, the selected biomarkers might be considered for implementation in routine clinical diagnosis to refine prognostic modelling and to help patient stratification in clinical decision-making.

## Figures and Tables

**Figure 1 cancers-15-00597-f001:**
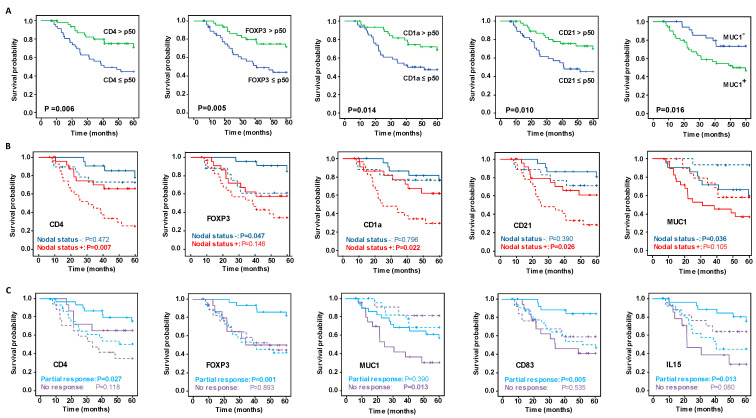
Kaplan–Meier analysis of five-year survival probability with respect to significant immunohistochemistry (IHC) and mRNA marker expression. (**A**) Kaplan–Meier analysis of overall survival (OS) among all patients. (**B**) Superimposed Kaplan–Meier curves of survival probability stratified by nodal status at diagnosis, and (**C**) depending on the type of response to neoadjuvant chemotherapy. In the plots in B and C, the continuous lines show the OS of patients with IHC immune marker concentrations greater than the median of all patients or when mRNA markers were expressed, and the dashed lines show the OS when IHC immune marker concentrations were less than the median of all patients or when the mRNA markers were not expressed. Significance of the log-rank test is indicated in the figures, with significant differences between variables indicated by probabilities highlighted in bold.

**Figure 2 cancers-15-00597-f002:**
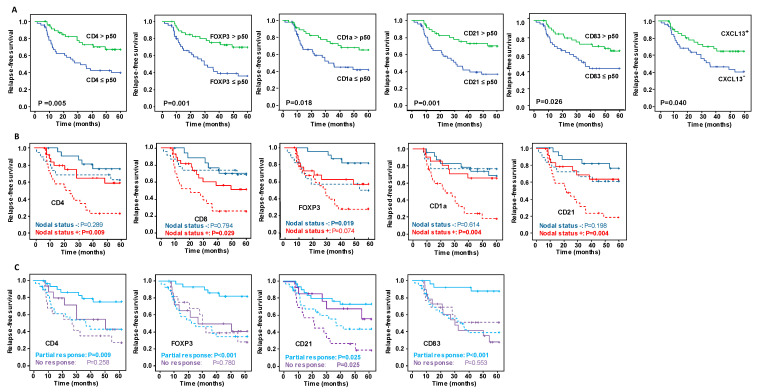
Kaplan–Meier analysis of five-year relapse-free survival (FRS) with respect to significant immunohistochemistry (IHC) and mRNA markers expression. (**A**) Kaplan–Meier analysis of RFS among all patients. (**B**) Superimposed Kaplan–Meier curves of RFS stratified by nodal status at diagnosis, and (**C**) depending on the type of response after NAC. In the plots in B and C, the continuous lines show the RFS of patients with IHC immune marker concentrations greater than the median of all patients or when mRNA markers were expressed, and the dashed lines show the RFS when IHC immune markers concentrations were less than the median of all patients or when mRNA markers were not expressed. Significance of the log-rank test is indicated in the figures, with significant differences between variables indicated by probabilities highlighted in bold.

**Figure 3 cancers-15-00597-f003:**
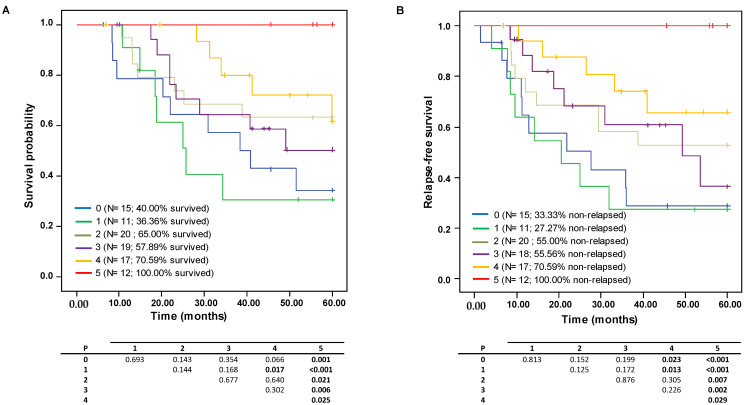
Kaplan–Meyer analysis of five-year survival probability with respect to the effect of the combinations having elevated content of 0 to 5 of the selected immune markers (CD4, FOXP3, CD1a, CD21, CD83) on (**A**) overall survival (OS) and (**B**) relapse-free survival (RFS) probabilities.

**Figure 4 cancers-15-00597-f004:**
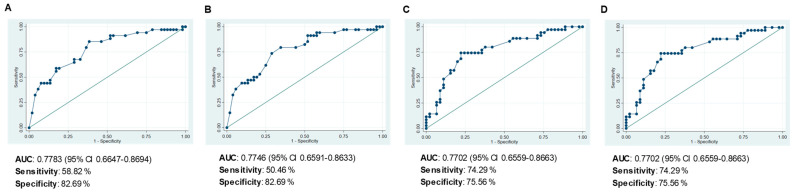
Receiver operating characteristic (ROC) curves evaluating the capacity of the final multivariate models to predict: (**A**) overall survival (OS), (**B**) OS after bootstrapping, (**C**) relapse-free survival (RFS), and (**D**) RFS after bootstrapping.

**Table 1 cancers-15-00597-t001:** Triple-negative breast cancer (TNBC) patient characteristics, globally and by survival and relapse categories during the five-year follow-up.

	All Patients (n = 96)	Non-Relapsed (n = 54)	Relapsed (n = 41)	*p*	Alive (n = 59)	Dead (n = 37)	*p*
Age Years, median (IQR)	51 (26)	51 (26)	52 (28)	0.473*	51 (26)	55 (28)	0.660 *
Menopausal status Pre-menopausal Menopausal	35 (44.9%) 43 (55.1%)	20 (45.5%) 24 (54.5%)	15 (45.5%) 18 (54.5%)	1.000 ^Ɨ^	21 (44.7%) 26 (55.3%)	14 (45.2%) 17 (54.8%)	1.000 ^Ɨ^
Histological grade <3 =3	17 (18.5%) 75 (81.5%)	9 (17.6%) 42 (82.4%)	8 (20.0%) 32 (80.0%)	0.988 ^Ɨ^	13 (23.2%) 43 (76.8%)	4 (11.1%) 32 (88.9%)	0.236 ^Ɨ^
Tumor diameter mm, median (IQR)	25.00 (22.00)	22.00 (25.00)	30.00 (25.00)	**0.011 ***	22.00 (23.00)	30.00 (20.00)	**0.021 ***
Nodal status Negative Positive	41 (43.6%) 53 (56.4%)	29 (55.8%) 23 (44.2%)	12 (29.3%) 29 (70.7%)	**0.019 ^Ɨ^**	32 (56.1%) 25 (43.9%)	9 (24.3%) 28 (75.7%)	**0.005 ^Ɨ^**
Ki67 degree ≤30% >30%	24 (27.9%) 62 (72.1%)	10 (20.0%) 40 (80.0%)	14 (40.0%) 21 (60.0%)	0.077 ^Ɨ^	14 (26.4%) 39 (73.6%)	10 (30.3%) 23 (69.7%)	0.886 ^Ɨ^
Pathological response Partial response Without response	63 (65.6%) 33 (34.4%)	40 (74.1%) 14 (25.9%)	23 (56.1%) 18 (43.9%)	0.106 ^Ɨ^	42 (71.2%) 17 (28.8%)	21 (56.8%) 16 (43.2%)	0.219 ^Ɨ^
Surgery Tumorectomy Mastectomy No lymphadenectomy Lymphadenectomy	36 (38.3%) 58 (61.7%) 20 (21.3%) 74 (78.7%)	21 (40.4%) 31 (59.6%) 13 (25.0%) 39 (75.0%)	14 (34.1%) 27 (65.9%) 7 (17.1%) 34 (82.9%)	0.688 ^Ɨ^ 0.503 ^Ɨ^	25 (43.9%) 32 (56.1%) 14 (24.6%) 43 (75.4%)	11 (29.7%) 26 (70.3%) 6 (16.2%) 31 (83.8%)	0.246 ^Ɨ^ 0.479 ^Ɨ^
Adjuvant chemotherapy No Yes	77 (82.8%) 16 (17.2%)	42 (80.8%) 10 (19.2%)	35 (87.5%) 5 (12.5%)	0.561 ^Ɨ^	47 (82.5%) 10 (17.5%)	30 (83.3%) 6 (16.7%)	1.000 ^Ɨ^
Adjuvant radiotherapy No Yes	4 (4.5%) 84 (95.5%)	3 (6.0%) 47 (94.0%)	1 (2.6%) 37 (97.4%)	0.631 ^Ɨ^	3 (5.5%) 52 (94.5%)	1 (3.0%) 32 (97.0%)	1.000 ^Ɨ^
Relapse No Yes	54 (56.8%) 41 (43.2%)	-	-	-	52 (88.1%) 7 (11.9%)	2 (5.6%) 34 (94.4%)	**<0.001 ^Ɨ^**
Overall survival Months, median (IQR)	47.32 (38.10)	60.00 (14.60)	25.07 (25.65)	**<0.001 ***	-	-	-
Relapse-free survival Months, median (IQR)	40.67 (46.89)	-	-	-	60.00 (16.13)	13.73 (14.88)	**<0.001 ***
Survival status Alive Dead	59 (61.5%) 37 (38.5%)	52 (96.3%) 2 (3.7%)	7 (17.1%) 34 (82.9%)	**<0.001 ^Ɨ^**	-	-	-

Data are presented as the median (interquartile range) for the Mann–Whitney U test *, and the number of patients (percentage) in each category for the Chi-squared test ^Ɨ^. The variables of age, menopausal status, histological grade, tumor diameter, nodal status, Ki67 degree were recorded at the time of diagnosis. Overall survival and relapse-free survival were censored at 60 months.

**Table 2 cancers-15-00597-t002:** Cellular and genetic immune markers in the residual tumor microenvironment according to survival and relapse during the five-year follow-up.

	Non-Relapsed (n = 54)	Relapsed (n = 41)	*p*	Alive (n = 59)	Death (n = 37)	*p*
CD4^+^ T lymphocytes	0.85 (2.29)	0.28 (0.97)	**0.004 ***	0.86 (2.33)	0.30 (0.74)	**0.004 ***
CD8^+^ T lymphocytes	0.81 (2.08)	0.37 (0.89)	0.051 *	0.79 (1.95)	0.37 (0.80)	0.055 *
FOXP3^+^ regulatory T cells	0.06 (0.19)	0.03 (0.06)	**0.004 ***	0.06 (0.19)	0.03 (0.06)	**0.020 ***
CD57^+^ NK cells	0.02 (0.06)	0.05 (0.39)	0.093 *	0.03 (0.09)	0.03 (0.32)	0.698 *
CD68^+^ macrophages	1.72 (2.53)	1.42 (2.55)	0.604 *	1.91 (3.48)	1.39 (2.35)	0.418 *
CD1a^+^ dendritic cells	0.12 (0.43)	0.07 (0.18)	**0.032 ***	0.12 (0.41)	0.07 (0.18)	0.053 *
CD21^+^ dendritic cells	0.001 (0.009)	0.000 (0.002)	**0.002 ***	0.001 (0.007)	0.000 (0.002)	**0.013 ***
CD83^+^ dendritic cells	0.07 (0.18)	0.03 (0.11)	0.103 *	0.06 (0.17)	0.03 (0.11)	0.201 *
CD15^+^ granulocytes	1.05 (4.60)	2.27 (5.28)	0.523 *	1.17 (4.74)	1.51 (4.53)	0.836 *
HLA-DR^+^ APC	14.94 (28.17)	9.63 (12.32)	0.101 *	14.86 (28.64)	9.63 (11.35)	0.156 *
CD31^+^ endothelial cells	1.35 (2.81)	1.18 (1.90)	0.173 *	1.37 (2.81)	1.15 (1.73)	0.114 *
CD34^+^ endothelial cells	1.88 (1.69)	1.90 (1.61)	0.921 *	2.11 (1.87)	1.86 (1.16)	0.407 *
CD138^+^ cells	26.25 (34.49)	25.76 (37.63)	0.940 *	23.84 (35.42)	27.49 (38.88)	0.454 *
**CXCL13** Absence Presence	20 (40.8%) 29 (59.2%)	25 (64.1%) 14 (35.9%)	**0.050 ^Ɨ^**	24 (44.4%) 30 (55.6%)	22 (62.9%) 13 (37.1%)	0.139 ^Ɨ^
**IL6** Absence Presence	16 (33.3%) 32 (66.7%)	14 (36.8%) 24 (63.2%)	0.911 ^Ɨ^	18 (34.0%) 35 (66.0%)	12 (35.3%) 22 (64.7%)	1.000 ^Ɨ^
**IL10** Absence Presence	42 (85.7%) 7 (14.3%)	35 (89.7%) 4 (10.3%)	0.748 ^Ɨ^	46 (85.2%) 8 (14.8%)	32 (91.4%) 3 (8.6%)	0.516 ^Ɨ^
**IL15** Absence Presence	22 (46.8%) 25 (53.2%)	22 (57.9%) 16 (42.1%)	0.424 ^Ɨ^	25 (48.1%) 27 (51.9%)	19 (55.9%) 15 (44.1%)	0.626 ^Ɨ^
**MMP1** Absence Presence	29 (59.2%) 20 (40.8%)	27 (71.1%) 11 (28.9%)	0.357 ^Ɨ^	35 (64.8%) 19 (35.2%)	22 (64.7%) 12 (35.3%)	1.000 ^Ɨ^
**MMP9** Absence Presence	14 (29.2%) 34 (70.8%)	15 (39.5%) 23 (60.5%)	0.439 ^Ɨ^	17 (32.1%) 36 (67.9%)	12 (35.3%) 22 (64.7%)	0.938 ^Ɨ^
**MMP12** Absence Presence	36 (78.3%) 10 (21.7%)	34 (89.5%) 4 (10.5%)	0.281 ^Ɨ^	41 (80.4%) 10 (19.6%)	30 (88.2%) 4 (11.8%)	0.511 ^Ɨ^
**MUC1** Absence Presence	23 (46.9%) 26 (53.1%)	13 (34.2%) 25 (65.8%)	0.329 ^Ɨ^	27 (50.0%) 27 (50.0%)	9 (26.5%) 25 (73.5%)	**0.050 ^Ɨ^**
**TNF-α** Absence Presence	30 (57.7%) 22 (42.3%)	22 (57.9%) 16 (42.1%)	1.000 ^Ɨ^	33 (57.9%) 24 (42.1%)	19 (55.9%) 15 (44.1%)	1.000 ^Ɨ^

For immunohistochemically stained markers, data are presented as the median (interquartile range) of the percentage of positive-stained area for the Mann–Whitney U test * and for mRNA expression levels, data (negative or positive) are presented as the number (percentage) of patients in each category for the chi-squared ^Ɨ^ test.

**Table 3 cancers-15-00597-t003:** Univariate and multivariate Cox analyses of markers and clinico-pathological variables associated with survival and relapse during the five-year follow-up in TNBC patients.

	Univariate Analysis		Multivariate Analysis		Multivariate Analysis after Bootstraping	
**Variables Associated with OS**	**HR (95% CI)**	* **p** *	**HR (95% CI)**	* **p** *	**HR (95% CI)**	* **p** *
Tumor diameter	1.018 (1.004–1.032)	0.009	-	-	-	-
Nodal status at baseline Positive Negative	3.154 (1.485–6.699) 1.0	**0.003**	4.061 (1.653–9.973) 1.0	**0.002**	2.869 (1.335–6.168) 1.0	**0.007**
CD4	0.746 (0.566–0.983)	**0.038**	0.642 (0.435–0.950)	**0.027**	0.773 (0.601–0.994)	0.083
CD8	0.816 (0.651–1.024)	0.079	-	-	-	-
FOXP3	0.012 (0.000–0.583)	**0.026**	-	-	-	-
HLA-DR	0.981 (0.962–1.000)	**0.049**	-	-	-	-
CD31	0.793 (0.631–0.997)	**0.047**	-	-	-	-
MUC1 Presence Absence	2.472 (1.153–5.301) 1.0	**0.020**	2.296 (1.049–5.026) 1.0	**0.038**	2.655 (1.237–5.697) 1.0	**0.006**
**Variables associated with RFS**	**HR (95% CI)**	* **p** *	**HR (95% CI)**	* **p** *	**HR (95% CI)**	* **p** *
Tumor diameter	1.018 (1.006–1.031)	**0.004**	1.014 (1.001–1.028)	**0.036**	1.014 (1.001–1.028)	**0.036**
Nodal status at baseline Positive Negative	2.509 (1.276–4.933) 1.0	**0.008**	2.749 (1.270–5.954) 1.0	**0.010**	2.749 (1.270–5.954) 1.0	**0.008**
Response Partial Without	1.766 (0.951–3.279) 1.0	0.072	-	-	-	-
CD4	0.792 (0.633–0.992)	**0.042**	-	-	-	-
CD8	0.815 (0.660–1.007)	0.058	-	-	-	-
FOXP3	0.007 (0.000–0.334)	**0.012**	-	-	-	-
CD83	0.120 (0.010–1.437)	0.094	-	-	-	-
CD15	1.042 (0.993–1.094)	0.097	-	-	-	-
HLA-DR	0.982 (0.965–0.999)	**0.043**	-	-	-	-
CD31	0.820 (0.673–0.999)	**0.049**	-	-	-	-
CXCL13 Presence Absence	0.510 (0.265–0.982) 1.0	**0.044**	0.453 (0.220–0.933) 1.0	**0.032**	0.453 (0.220–0.933) 1.0	**0.028**

## Data Availability

The raw data supporting the conclusions of this article are available from the corresponding authors upon reasonable request.

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
