# Peer review of "Prognostic Implications of the Residual Tumor Microenvironment after Neoadjuvant Chemotherapy in Triple-Negative Breast Cancer Patients without Pathological Complete Response"

_cancers, 2023, doi:10.3390/cancers15030597_

Round 1

Reviewer 1 Report

Comments and suggestions for authors

The manuscript entitled “Prognostic implications of the residual tumor microenvironment after neoadjuvant chemotherapy in triple-negative breast cancer patients without pathological complete response ”. 

The authors represented the combining predictive biomarkers of survival and relapse in TNBC patients with promising results. However, the reviewer notes some critical aspects.

1.     Please indicate the criteria to quantify the percentage of positive-stained area of each marker staining (line 201). It can be included in supplementary files. 

2.     When poor staining occur, authors decided to used only single cylinder staining (line 205). How does single staining represent the whole tumor tissue as we known that TNBC is a heterogenous disease?

3.     Please correct font size in text (line 290-293).

Author Response

Please see attached the manuscript revised.

We sincerely appreciate the revision of our manuscript and the feedback given by the reviewers, which undoubtedly substantially improve the quality of the manuscript. We have carefully revised the manuscript according to the referees' comments and provided a point-by-point response to the reviewers' suggestions and comments below. Moreover, we have checked that all references are relevant to the contents of the manuscript, and we proposed to replace reference 49 with: Lopez, C.; Callau, C.; Bosch, R.; Korzynska, A.; Jaen, J.; Garcia-Rojo, M.; Bueno, G.; Salvado, M. T.; Alvaro, T.; Onos, M. et al. Development of automated quantification methodologies of immunohistochemical markers to determine patterns of immune response in breast cancer: a retrospective cohort study. BMJ Open. 2014, 4:e005643. We have also marked any change to the manuscript using track changes. Finally, we have checked and confirmed the authorship and, we are preparing the attached manuscript signature to send to your editorial office.

We have found mirror mistakes in supplementary files, on captions S6 and S7, which we would like to correct.

Fig S6. Superimposed Kaplan–Meier curves of five-year overall survival (OS) by tumor microenvironment (TME) markers by type of response. The continuous lines show the OS of patients with immunohistochemistry (IHC) immune markers concentrations higher than the median, or with positive mRNA expression levels for markers determined by in situ hybridization (ISH). The dashed lines show the RFSOS when IHC immune markers concentrations were lower than the median or when mRNA was not expressed for ISH markers. Significance levels for the log-rank test are indicated in the figures.

Fig S7. Superimposed Kaplan–Meier curves of five-year relapse-free survival (RFS) by tumor microenvironment (TME) markers by type of response. The continuous lines show the OSRFS of patients with immunohistochemistry (IHC) immune markers concentrations higher than the median, or with positive mRNA expression levels for markers determined by in situ hybridization (ISH). The dashed lines show the RFS when IHC immune markers concentrations were lower than the median or when mRNA was not expressed for ISH markers. Significance levels for the log-rank test are indicated in the figures.

We look forward to receiving your response at your earliest convenience and thank you in advance for considering our manuscript.

Yours sincerely,

Marylène Lejeune

Department of Pathology, Oncological Pathology and Bioinformatics Research Group

Hospital de Tortosa Verge de la Cinta

C/Esplanetes 14

Tortosa 43500. Spain

Phone/Fax: +34 977519104

Response to reviewers:

Reviewer 1

The manuscript entitled “Prognostic implications of the residual tumor microenvironment after neoadjuvant chemotherapy in triple-negative breast cancer patients without pathological complete response”. The authors represented the combining predictive biomarkers of survival and relapse in TNBC patients with promising results. However, the reviewer notes some critical aspects.

  1. Please indicate the criteria to quantify the percentage of positive-stained area of each marker staining (line 201). It can be included in supplementary files. 

The quantification procedure is an automated image analysis consisting of two steps: the evaluation of the total area of each cylinder and the evaluation of the area of each cylinder that was positively stained with the targeted marker (lines 200-205). Since the entire procedure was detailed in our previous studies cited in the manuscript (Lopez, C et al. 2014 and Callau, C. et al 2015), we did not think it necessary to include it in the supplementary files.

However, in the present study, we used the percentage of positive-stained area of each marker in the cylinders and not the number of specific positive stained cells, due to the large variability of the cells counts, which arises from the large number of variables that influence the cell count, such as variation in cell size, shape, and distribution (clusters). In addition, the subcellular location of DAB-staining (membrane, cytoplasm, nucleus) also makes it difficult to determine the positive cells, as it requires the application of different filters and/or other procedures to unify the positive pixels and thus determine a number of positive cells (Lejeune, M et al. 2008).

References

Lopez, C.; Callau, C.; Bosch, R.; Korzynska, A.; Jaen, J.; Garcia-Rojo, M.; Bueno, G.; Salvado, M. T.; Alvaro, T.; Onos, M. et al. Development of automated quantification methodologies of immunohistochemical markers to determine patterns of immune response in breast cancer: a retrospective cohort study. BMJ Open. 2014, 4:e005643.

Callau, C.; Lejeune, M.; Korzynska, A.; Garcia, M.; Bueno, G.; Bosch, R.; Jaen, J.; Orero, G.; Salvado, T. and Lopez, C. Evaluation of cytokeratin-19 in breast cancer tissue samples: a comparison of automatic and manual evaluations of scanned tissue microarray cylinders. Biomed Eng Online. 2015, 14 Suppl 2, S2.

Lejeune, M.; Jaen, J.; Pons, L.; Lopez, C.; Salvado, M. T.; Bosch, R.; Garcia, M.; Escriva, P.; Baucells, J.; Cugat, X.; et al. Quantification of diverse subcellular immunohistochemical markers with clinicobiological relevancies: validation of a new computer-assisted image analysis procedure. J Anat. 2008, 212, 868-878.

  1. When poor staining occur, authors decided only single cylinder staining (line 205). How does single staining represent the whole tumor tissue as we known that TNBC is a heterogenous disease?

We agree with the reviewer that considering only one single cylinder staining may not always represent the whole tumour tissue so that the selection of the tissue area represents a fundamental factor for the evaluation of immune markers. In this work, pathologists first evaluate the area with residual tumor with the best representation of TME components to be included in the TMA (tumour and infiltrating stroma) and further reviewed all immunostaining performed for each cylinder of the tissue microarrays.

However, problems might occur during the slides preparation and some cylinder might partially degrade throughout the process. In our study, this only occurs in 3.1% to 9.4% of cases depending on the marker. Therefore, due to the selection of representative area, the consideration of a single cylinder in less than 10% of the cases would not invalidate the overall results of our study. Accordingly, we have reflected these percentages in lines 206-208.

  1. Please correct font size in text (line 290-293). Font size has been corrected.

Reviewer 2 Report

This is a study that aims to investigate the linkage between the residual tumor microenvironment after neoadjuvant chemotherapy in predicting survival. The investigators identify immune markers that are linked with survival. The study is generally well conducted and interesting.

I have one major issue with this otherwise well-done study: the distribution and composition of immune cells within TNBA and at the leading edge is typically quite different. Significant immune markers should undergo a detailed analysis to determine if these marker changes occur at both the leading edge and/or the tumor interior. 

Author Response

We sincerely appreciate the revision of our manuscript and the feedback given by the reviewers, which undoubtedly substantially improve the quality of the manuscript. We have carefully revised the manuscript according to the referees' comments and provided a point-by-point response to the reviewers' suggestions and comments below. Moreover, we have checked that all references are relevant to the contents of the manuscript, and we proposed to replace reference 49 with: Lopez, C.; Callau, C.; Bosch, R.; Korzynska, A.; Jaen, J.; Garcia-Rojo, M.; Bueno, G.; Salvado, M. T.; Alvaro, T.; Onos, M. et al. Development of automated quantification methodologies of immunohistochemical markers to determine patterns of immune response in breast cancer: a retrospective cohort study. BMJ Open. 2014, 4:e005643. We have also marked any change to the manuscript using track changes. Finally, we have checked and confirmed the authorship and, we are preparing the attached manuscript signature to send to your editorial office.

We have found mirror mistakes in supplementary files, on captions S6 and S7, which we would like to correct.

Fig S6. Superimposed Kaplan–Meier curves of five-year overall survival (OS) by tumor microenvironment (TME) markers by type of response. The continuous lines show the OS of patients with immunohistochemistry (IHC) immune markers concentrations higher than the median, or with positive mRNA expression levels for markers determined by in situ hybridization (ISH). The dashed lines show the RFSOS when IHC immune markers concentrations were lower than the median or when mRNA was not expressed for ISH markers. Significance levels for the log-rank test are indicated in the figures.

Fig S7. Superimposed Kaplan–Meier curves of five-year relapse-free survival (RFS) by tumor microenvironment (TME) markers by type of response. The continuous lines show the OSRFS of patients with immunohistochemistry (IHC) immune markers concentrations higher than the median, or with positive mRNA expression levels for markers determined by in situ hybridization (ISH). The dashed lines show the RFS when IHC immune markers concentrations were lower than the median or when mRNA was not expressed for ISH markers. Significance levels for the log-rank test are indicated in the figures.

We look forward to receiving your response at your earliest convenience and thank you in advance for considering our manuscript.

Yours sincerely,

Marylène Lejeune

Department of Pathology, Oncological Pathology and Bioinformatics Research Group

Hospital de Tortosa Verge de la Cinta

C/Esplanetes 14

Tortosa 43500. Spain

Phone/Fax: +34 977519104

Reviewer 2

This is a study that aims to investigate the linkage between the residual tumor microenvironment after neoadjuvant chemotherapy in predicting survival. The investigators identify immune markers that are linked with survival. The study is generally well conducted and interesting.

We are grateful to the reviewer for the great considerations given to our manuscript. 

I have one major issue with this otherwise well-done study: the distribution and composition of immune cells within TNBA and at the leading edge is typically quite different. Significant immune markers should undergo a detailed analysis to determine if these marker changes occur at both the leading edge and/or the tumor interior. 

It is true that there is heterogeneity within the TME of the TNBC phenotype, and immune infiltrate such as tumour-infiltrating lymphocytes (TILs) maybe present in different concentrations in the intra- and peritumoral stroma. However, most of the studies have demonstrated the predictive and prognostic value of intratumoral immune infiltrates in breast cancer patients and only a few works have evaluated the invasive border in breast cancer. The later have shown that the presence of TIL in the peritumoral area does not correlate with clinical factors (Acs et al., 2017; Al-Saleh et al., 2017) or with the presence of ALN metastasis at diagnosis (López, C. et al. 2020). In our previous work, the immune infiltrates in the peritumoral area in BC did not show prognostic implications, at least for 11 immune markers (CD4, CD8, CD21, CD68, CD123, LAMP3, CD57, CD1a, CD83, S100 and FOXP3) (López, C. et al. 2020).

In the present study, the area of the biopsies included in the TMA corresponds to the residual tumor after receiving NAC but not to the entire intratumoral area of biopies obtained at diagnosis as in the referenced studies. The pathologists have carefully selected these areas with the best representation of the whole-tissue sample (tumour and infiltrating stroma) and all the immunostains performed for each cylinder of the tissue microarrays have been subsequently revised.

In any case, we will take into consideration the evaluation of the peritumoral area in future studies. Accordingly, this idea has been added as future perspectives at the end of the discussion (lines 558-560).

References

Acs B, Madaras L, Tokes AM, Kovacs AK, Kovacs E, Ozsvari-Vidakovich M, Karaszi A, Birtalan E, Dank M, Szasz AM, Kulka J. 2017. PD-1, PD-L1 and CTLA-4 in pregnancy-related_and in early-onset breast cancer: a comparative study. Breast 35:69_77

Al-Saleh K, Abd El-Aziz N, Ali A, Abozeed W, Abd El-Warith A, Ibraheem A, Ansari J, Al-Rikabi A, Husain S, Nabholtz JM. 2017. Predictive and prognostic significance of CD8 (+) tumor-infiltrating lymphocytes in patients with luminal B/HER 2 negative breast cancer treated with neoadjuvant chemotherapy. Oncology Letters 14:337 344

López C, et al. Peritumoral immune infiltrates in primary tumours are not associated with the presence of axillary lymph node metastasis in breast cancer: a retrospective cohort study. PeerJ 8:e9779 http://doi.org/10.7717/peerj.9779.

Round 2

Reviewer 2 Report

The investigators have responded to the concerns. Accept in present form.